# Effect of the Notch-to-Depth Ratio on the Post-Cracking Behavior of Steel-Fiber-Reinforced Concrete

**DOI:** 10.3390/ma14020445

**Published:** 2021-01-18

**Authors:** José Valdez Aguilar, César A. Juárez-Alvarado, José M. Mendoza-Rangel, Bernardo T. Terán-Torres

**Affiliations:** Facultad de Ingeniería Civil (FIC-UANL), Universidad Autónoma de Nuevo León, Av. Universidad S/N, Cd. Universitaria, San Nicolás de los Garza 66455, Mexico; abelardo.valdezgr@uanl.edu.mx (J.V.A.); jose.mendozarn@uanl.edu.mx (J.M.M.-R.); bernardo.terantrr@uanl.edu.mx (B.T.T.-T.)

**Keywords:** steel-fiber-reinforced concrete, hook-end steel fiber, post-cracking behavior, notch-to-depth ratio, ductility, toughness, fracture energy

## Abstract

Concrete barely possesses tensile strength, and it is susceptible to cracking, which leads to a reduction of its service life. Consequently, it is significant to find a complementary material that helps alleviate these drawbacks. The aim of this research was to determine analytically and experimentally the effect of the addition of the steel fibers on the performance of the post-cracking stage on fiber-reinforced concrete, by studying four notch-to-depth ratios of 0, 0.08, 0.16, and 0.33. This was evaluated through 72 bending tests, using plain concrete (control) and fiber-reinforced concrete with volume fibers of 0.25% and 0.50%. Results showed that the specimens with a notch-to-depth ratio up to 0.33 are capable of attaining a hardening behavior. The study concludes that the increase in the dosage leads to an improvement in the residual performance, even though an increase in the notch-to-depth ratio has also occurred.

## 1. Introduction

Concrete is the most used construction material around the world due to its versatility, formability, and the widespread availability of its ingredients [1,2,3]. However, concrete barely possesses tensile strength, and it is susceptible to cracking, which leads to a reduction of its service life, since, once cracks start developing, concrete lacks mechanical strength and fails suddenly [4,5,6,7,8].

Consequently, material science and concrete technology have strived to find a complementary material that allows for alleviating these drawbacks and, once the first-crack load is attained, to provide the ability to prevent complete fracture. In other words, they are striving to find a material that favors the improvement on strain capacity and energy absorption.

One of these materials is steel fibers. Several studies with steel fibers have proven a minimum improvement in the concrete compressive strength [9,10,11,12,13]. However, a large influence in performance has been attained on tensile strength tests [14,15,16], and even more significant on impact loading and residual strength tests [17,18,19,20].

Steel fibers are suitable for reinforcing concrete since they provide an energy dissipation mechanism and control the crack propagation in residual stages more efficiently than plain concrete. This latter is one of the main advantages in the use of this reinforcing material. However, other characteristics can be affected due to the effective concrete area, for instance, the residual performance of the fibers, which is one of the primary factors in the transmissibility of stresses through the bond developed between the fiber and the cementitious matrix [21].

The study of the residual performance is possible through notched specimens with controlled-induced crack, from where representative data are obtained, and for which the dispersion coefficients are less than those for un-notched specimens [22,23].

Several advantages have been reported on the use of steel-fiber-reinforced concrete (SFRC), for instance, the reduction in terms of costs and an improvement in the quality of structures, mainly in crack control and under cyclic loading [24]. Some authors have focused on the tension stiffening effect on reinforcing bars embedded in plain concrete and steel-fiber-reinforced concrete ties, varying the main parameters, such as the fiber volume, the maximum aggregate size, and the diameter of the steel reinforcement, observing the effectiveness of steel-fiber-reinforced concrete in controlling the crack pattern of reinforced concrete structures [25]. On the other hand, it has been reported that the flexural crack width is significantly reduced with the addition of steel fibers and that the first cracking load and maximum load are increased with the addition of steel fibers [26]. Other studies with other kinds of fibers, i.e., polypropylene fibers (PFRC), have found that the fracture energy is higher because of the strong dependency between PFRC post-cracking performance and fiber distribution and orientation [27].

Regarding bending tests for characterizing fiber-reinforced concrete, it has been found that the failure mode of the specimen is influenced by its span-to-depth ratio; this means that its failure mode is governed by shear or bending [28].

For the residual strength assessment, international codes and standards have proposed three-point bending tests on beam specimens notched at midspan, to control the crack development, by establishing a notch-to-depth ratio (i.e., a/d) of 0.16 [20].

In this study, the effect of the notch-to-depth ratio in the fiber-reinforced concrete was studied, by utilizing two dosages of steel fibers added, one of 20 kg/m^3^ (0.25%) (Series 1) and another of 40 kg/m^3^ (0.50%) (Series 2), in specimens subjected to flexural tension (i.e., three-points bending test) on prismatic specimens of 150 mm × 150 mm × 600 mm. The crack initiation was controlled by inducing a notch in the specimens [15,17,18,29] of 13, 25, and 50 mm, which led to a notch-to-depth ratio (a/d) of 0.08, 0.16, and 0.33, respectively. The purpose of the herein research was to study the effect on the residual performance of the steel fibers, the reduction of the concrete area through the computation of the characteristic residual stresses, and the classification of residual strength to determine the strength provided by steel fibers by the enlargement and the propagation of the macrocrack toward the reduced concrete areas.

The novelty of the present research lies in the determination of the residual performance of the fiber-reinforced concrete for notch-to-depth ratios (a/d) lower than 0.16; this differs from other studies found in the literature where larger notch-to-depth ratios are considered. The consideration of lower ratios allows for the study of the post-cracking behavior and the fracture energy in a fracture process zone larger than that of the standards.

## 2. Theoretical Aspects

To understand the influence of the (a/d) ratio, it is necessary to mention some of the concepts of nonlinear fracture mechanics in concrete. In general terms, fracture mechanics is defined as a failure theory of great utility, since it utilizes energetic criteria that, together with strength criteria, account for the propagation of the crack through the structures [30]. However, despite its utility, it has been found out that the behavior of the concrete is not defined through the linear elastic fracture mechanics (LEFM), since concrete develops a fracture process zone (FPZ) relatively larger, which endures progressive damage due to softening caused by microcracking. This leads to a reduction in energy flow released at the tip of the crack; at the same time, the combined surface area of cracking increases, which improves the capacity of the energy absorption at the fracture process zone (FPZ), and thus, to understand the behavior of concrete, it is essential to elaborate on nonlinear fracture mechanics [30].

The nonlinear fracture mechanics is one of the theories that better describes the concrete behavior; nonlinear fracture mechanics is the study of the cracking of solid bodies that show a nonlinear constitutive response in nature, contrary to LEFM, where geometrical and material linearity is considered [31].

The construction materials can show several behaviors depending on the applied loads; some materials can manifest barely or null deformation capacity, so they are considered to be fragile materials. On the contrary, there exist materials considered to be ductile. Concrete is a special case, as its behavior cannot be characterized completely as fragile, since it is more likely described as a quasi-fragile material. Concrete shows a gradual decay of the tensile stress (strain softening) with or without an improvement on the tensile strength, after the development of the first crack (strain hardening); hence, in general terms, the failure can be presented without yielding [32,33].

Another important aspect to consider is the capacity of energy absorption, which is obtained as the area under the curves of load-displacement or load-crack opening. The consideration of energy absorption is necessary, especially under dynamic loading, since it determines the ductility of the structure. In fragile materials, the elastic energies are dissipated as superficial energy, without an FPZ. Meanwhile, in ductile materials, the FPZ is a plastic zone that can dissipate a large amount of energy, larger than the superficial energy; for quasi-fragile materials, such as concrete, the FPZ is usually larger than the zone for ductile or fragile materials [34,35] and dissipates an important amount of energy before the failure, which provides a post-cracking nonlinear response (softening) [32].

The addition of steel fibers to the concrete, in a determined fraction of volume, improves ductility and increases the initial width of the FPZ, which gives, as a result, the enlargement of the zone due to the extraction of the fibers [36]. The steel fibers, randomly distributed, show their most important effect after the cracking of the matrix, by delaying the formation of cracks, by limiting their growth, and by reducing the crack tip opening displacement, since the fibers suppress the cracks by means of a bridging mechanism during their extraction process [37,38].

In the same manner, the use of steel fibers largely increases the energy absorption and ductility [39]. It is important to mention that the concrete fragile behavior is proportional to the increase of its compressive strength, and the addition of steel fibers aids to withdraw such fragility induced, leading to the production of a material with improved behavior of tensile strength, ductility, and toughness [40,41].

In comparison with plain concrete, this material shows an extended softening branch that is characterized by a significant tensile residual strength and higher fracture energy [9], with the latter being a prime ingredient to measure the fracture process of quasi-fragile materials.

## 3. Materials and Methods

For this research, 4 plain concrete mixtures were produced, one for each notch-to-depth ratio used. Furthermore, 8 mixtures of fiber-reinforced concrete (4 for each fiber percentage used) were also produced. The mixtures were produced by using cement OPC 40, which satisfies NMX-C-414-ONNCCE-2014 [42]; standard crushed limestone as aggregate, with a maximum size of 19 mm (3/4″) and a fineness modulus of 2.42; water; and a polycarboxylate superplasticizer as additive.

The steel fibers utilized were “hooked-end” fibers, with a length (l_f_) of 50 mm, a diameter (d_f_) of 1 mm, an aspect ratio (l_f_/d_f_) of 50, and a tensile strength of 1130 MPa; the fiber volumes were 0.25% and 0.50%, which are the volumes recommended to obtain a residual strength according to the standard EN 14845-1, 2007 [43]. This standard also states a maximum cement content of 350 kg/m^3^ and a water/cement ratio of 0.55. The composition for each mixture is shown in Table 1.

### 3.1. Fracture Test Method

Three different mixtures were used, along with four different notch depths (i.e., 0, 13, 25, and 50 mm). The first mixture was required as the reference series (i.e., volume fiber, Vf = 0%). From this mixture, 6 beams of 150 mm × 150 mm × 600 mm for each depth were fabricated; thus, a total of 24 specimens were constructed.

In the second and third mixes, volumes of fibers (Vf) of 0.25% and 0.50% were used, respectively. Consequently, a total of 24 specimens were also constructed for each mix and for each notch depth. Hence, a total of 48 prismatic specimens were constructed with fibers, with the dimensions and notch depths aforementioned.

The effect of the addition of steel fibers on the post-cracking performance was evaluated in the 72 specimens (plain concrete and fiber-reinforced concrete), through a flexural tension test of the notched beams with notch depths previously described, and with their corresponding notch-to-depth ratio (a/d) of 0, 0.08, 0.16, and 0.33, respectively. It is worth mentioning that the notching procedure was carried out according to the standard EN 14651-2005, where the common notch depth is 25 mm [20].

In this experimental program, the crack-mouth-opening displacement method (CMOD) was used, plotted against the applied load. The measurements of the openings were conducted by employing clip-on-type extensometers, with a gauge length of 20 mm and a stroke of +12 mm/–2 mm (see Figure 1a and Figure 2a), with the objective of estimating the capacity to transfer stresses of the fibers through the cracking faces of the specimen by preventing the enlargement of the induced crack. In addition, a linear variable differential transformer (LVDT) with a linear range of 12.7 mm was placed at midspan of the specimen, to measure the displacement due to the applied load and to determine the contribution of the steel fibers in toughness and ductility of the composite material (see Figure 1b and Figure 2b).

### 3.2. Load and Stress at the Proportional Limit

Experimental results were obtained from the load at the proportional limit or from the load for the first crack (FL), which is the larger value of load recorded up to a CMOD of 0.05 mm [9,36]. The corresponding acting stress, at the occurrence of the first crack, was computed by means of Equation (1) [20].
(1)$$f_{L}=\frac{3F_{L}L}{2b{\left(\mathrm{hsp}\right)}^{2}}\;(N/{\mathrm{mm}}^{2})$$
where f_L_ = stress at the proportional limit (N/mm^2^), F_L_ = load at the proportional limit (N), L = span of the specimen (mm), b = width of the specimen (mm), and hsp = distance between the top face of the specimen and the tip of the induced crack (mm).

### 3.3. Normal and Characteristic Residual Stresses

The contribution of the steel fibers is of major importance at the residual stage. This contribution, at the post-cracking stage, is obtained through the normal stresses (f_Rj_), at each specific value of the measured CMOD, and computed by Equation (2) [20].
(2)$$f_{R,j}=\frac{3F_{R}L}{2b{\left(\mathrm{hsp}\right)}^{2}}\;(N/{\mathrm{mm}}^{2})$$
where f_R,j_ = normal residual stress at the point j (N/mm^2^), f_Rk,j_ = characteristic residual stress at point j (N/mm^2^), and F_R_ = load for a given crack-mouth opening displacement measured.

The values of f_R,1_, f_R,2_, f_R,3_, and f_R,4,_ are the normal residual stresses (N/mm^2^) for a 0.5, 1.5, 2.5, and 3.5 mm crack-mouth opening, respectively. In a similar manner, the values of f_Rk,1_, f_Rk,2_, f_Rk,3_, and f_Rk,4_ are the characteristic residual stresses (N/mm^2^) for a 0.5, 1.5, 2.5, and 3.5 mm crack-mouth opening, respectively.

In this research, the assessment of the characteristic residual strength was obtained according to the procedure established in CEB-FIP model code 2010 [44], through Equation (3), with the factors obtained in Molins and Arnau 2012; Rilem TC-162, 2003, as listed in Table 2 [45,46].
(3)$$f_{\mathrm{Rk},j}=f_{R,j}-k_{x}\;(N/{\mathrm{mm}}^{2})$$
where n = number of tested specimens, k_xN_ = statistical factor when the coefficient of variation of the population set is known, and k_xn_ = statistical factor when the coefficient of variation of the population set is unknown.

### 3.4. Fracture Energy

Two models were utilized in the estimation of the fracture energy in the fiber-reinforced concrete specimens, where their respective parameters and principles differ from each other. For model 1, proposed by Barros et al. [47], shown in Equation (4), parameters such as the mass of specimen and final displacement measured are needed. On the other hand, for the second model, the use of graphs is required [48].
(4)$$\mathrm{GF}=\frac{W_{f}+\;m\left(1-{\;a}^{2}\right){\mathrm{gS}}_{u\;}}{b\left(d-{\;a}_{0}\right)}\;(N-m)$$

Model 2, proposed by Kazemi et al. [48], is presented in Equation (5), and it assumes that the required work to fracture a specimen is proportional to the cracked surface:(5)$$\mathrm{GF}=\frac{1}{b}\left(\frac{{\mathrm{dW}}_{f}}{\mathrm{dr}}\right)\;(N-m)$$
where GF = total fracture energy (N/m); W_f_ = area under the curve load vs crack opening (or displacement), work due to fracture (N-m) or (J); m = mass of the specimen (kg); a = relation between the total length of the specimen (l) and span (length between supports) (L); $a_{0}$= initial notch depth (m); b = width of the specimen (mm); d = beam depth (m); g = gravity acceleration constant (9.8 m/s^2^); $S_{u\;}$= maximum value of displacement or crack opening measured (m); S = standard deviation of the set (N/mm^2^); L = length of the specimen (mm); and r = distance between the top face of the specimen and the crack tip (m).

## 4. Results and Discussion

### 4.1. Behavior at the Proportional Limit

Figure 3 shows the obtained results for the load at the proportional limit for each of the three series. It is observed that, as the notch-to-depth ratio (a/d) increases, the load capacity decreases, where reductions of the order of 33%, 53%, and 66% were obtained for the ratios (a/d) = 0.08, 0.16, and 0.33, respectively. Hence, as the notch-to-depth ratio increases, the concrete becomes prone to cracking failure. This behavior is due to the reduction of the fracture process zone (FPZ), also known as ligament length, which allows for a higher dissipation of energy during the cracking process [48]. This load reduction was also recognized by Zihai Shi [49], where the reduction of the ligament length led to a reduction of peak load.

In the same manner, in Figure 3, it can be also observed that the load resistance depends primarily on the concrete resistance area and not on the amount of fibers added. This can be recognized by observing the similar values of load attained for each ratio (a/d) at different fiber volumes, Vf, which indicates that, for a stage prior to the development of the first crack, the fibers provide barely or null influence on the strength of the composite material. From this behavior, a general graph (see Figure 4) was obtained, and its behavior can also be described by Equation (6).
(6)$$F_{L}\left(a/d\right)=-308,966{\left(\frac{a}{d}\right)}^{3}+372,190{\left(\frac{a}{d}\right)}^{2}-146,386\left(\frac{a}{d}\right)+28,621\;(N).$$

The required stresses for the first crack to appear, for each studied series, are presented in Table 3. In this table, the computed stresses tend to decrease as the notch depth increases. In addition, for those specimens without an initial induced notch, a higher stress was required for the first crack to appear, with respect to those with an initial notch. This behavior is consistent with a larger amount of concrete area presented in the specimen, in comparison to those with an initial induced notch. However, it is worth noting that the specimens with an (a/d) = 0 ratio showed the larger variation coefficient (CV) since the cracking process is not controlled and the crack may appear in different zones throughout the length of the specimen, leading to a variation in the residual behavior.

The least dispersion in the results was obtained for Series 1 and 2, with a ratio of (a/d) = 0.16, which is the notch-to-depth ratio suggested by the standard EN 14651, 2005 [20], whereby the best control in the cracking process was observed, in comparison with the remaining (a/d) ratios studied. Furthermore, with the ratio (a/d) = 0.16, a more representative behavior of the capability of the fibers was attained in the post-cracking stage. Moreover, at this ratio, the depth is large enough to favor the occurrence of the first crack in the desired zone and to generate a concrete area that is sufficiently large enough for the fibers to properly transfer stresses during the residual stage. 

### 4.2. Post-Cracking Behavior

Figure 5, Figure 6, Figure 7 and Figure 8 show the results obtained in the flexural tensile test for both series with steel fibers and for each (a/d) ratio. In these figures, it can be noticed that the load at the proportional limit (linear part of the curve) is independent of the amount of fibers used, as previously discussed. Furthermore, it can be also observed that the main effect of the fibers is obtained in the post-cracking stage [13,15,50].

In the figures corresponding to Series 2, a larger performance in the post-cracking stage is shown, in comparison with those corresponding to Series 1, reaching a residual load equal to, or even larger than, the average load obtained in the first crack-occurrence stage. This results in a hardening behavior due to the larger amount of fibers used (i.e., Vf = 0.50%), which improves the performance in the post-cracking stage by increasing the capacity to transfer stresses through the cracked faces.

By analyzing the results of both series of the obtained curves, in Figure 5, Figure 6, Figure 7 and Figure 8, the maximum performance of the specimens can be observed in the residual stage at values less than 4 mm of CMOD, which is equivalent to 3.44 mm of the displacement at the midspan, after which the post-cracking performance is reduced. This is of prime importance since this value is usually considered for computation of the fiber performance, which is obtained for 3.5 mm of CMOD or 3 mm of the deflection at midspan [9]

### 4.3. Normal Residual Stresses

In Figure 9, the results of the normal residual stresses (f_R,j_) are shown, for both studied series. Based on the (a/d) ratios utilized, it can be noticed that, from Figure 9a–d, the load at the proportional limit is consistent in value, independent of the amount of fibers used. This indicates that the fibers barely have an influence in the stage before the occurrence of the first crack. It is also worth noting that, by increasing the ratio (a/d), the residual performance developed is improved by the increase in the dosage of the fibers. For instance, the specimens with a ratio (a/d) = 0 manifested an increment in the residual performance of 61%. Meanwhile, an increment of 157%, 129%, and 86% was attained for the series with (a/d) ratios of 0.08, 0.16, and 0.33, respectively.

In the same manner, the maximum performance was reached with the increment on the dosage of fibers for the ratios between 0.08 and 0.16, while for 0.33, its residual performance decreased.

### 4.4. Characteristic Residual Stresses

In Figure 10a,b, the experimental results of the characteristic residual stresses (f_Rk,j_) for each (a/d) ratio, for both Series 1 and 2, are shown, along with their corresponding average stress curves for each case. It can be noticed that the characteristic residual stresses are significantly sensitive to the variation of the normal residual stresses; this can be verified in Figure 10a, where the ratios (a/d) = 0 and 0.08 showed a lower residual performance than those obtained for the ratios (a/d) = 0.16 and 0.33. Hence, the ratios (a/d) = 0 and 0.08 do not meet the minimum requirements established in the standard EN 14845-2, 2006 [41]. 

For Series 1, by increasing the ratio (a/d), a more suitable performance in the post-cracking stage was attained, this suggests that there is an adequate transmission of stresses on the faces of the specimens. On the other hand, in Figure 10b, by using a larger amount of fibers (Series 2), the characteristic residual stresses meet with the minimum requirements of the standard [41].

However, the behavior with respect to the ratios (a/d) of Series 2 is not consistent with what was obtained in Series 1, since the ratio (a/d) = 0.08 provides the best residual performance of all the ratios utilized (see Figure 10b). This can be the result of the influence of the amount of fibers, which can have better distribution in a larger area of concrete. For both series, the ratio (a/d) = 0 showed the poorest performance.

The assessment of the characteristic residual stress close to the average normal stress, obtained from experiments, requires a lower dispersion between tests. This can be attained from the increment on the number of specimens. This will result in a decrease in the statistical uncertainty factor, k_x_, and, thus, the variation in the results will be lower.

In Table 4, the classifications of the residual strength for both series are shown. According to the recommendations of the model code 2010 (MC-2010) [44,45], for the classification, factors k_xn_ = 2.33 and 2.18 were used, for the series where five and six specimens were tested, respectively. The (a/d) ratios that do not show any classification are those that did not attain the minimum characteristic residual strength of 1 N/mm^2^. 

The results obtained for the ratios (a/d) = 0 and 0.08 of Series 1 are low and insufficient in order to be able to reach the minimum characteristic residual stress and to be able to classify their residual behavior. On the contrary, the ratios (a/d) = 0.16 and 0.33 showed perfectly plastic behavior and a hardening behavior, in their respective residual response.

The results for Series 2, as shown in Table 4, indicate that, by increasing the (a/d) ratio, an increase in the performance in the residual stage will also occur, which can result in a hardening behavior, as presented on the (a/d) = 0.33 ratio. This proves that the fiber content is high enough to exhibit a hardening behavior under bending [50]; this suggests that, even though the peak load to reach the first crack decreases, by increasing the ratio (a/d), the addition of fibers can equate or even surpass such load, by improving the performance in the residual stage. This behavior will not depend on the amount of total fibers on the cracked surface, but rather on the amount of fibers that have an effective contribution in the control of the cracking process of the cement matrix.

### 4.5. Fracture Energy for the Steel-Fiber-Reinforced Concrete

The increase in the (a/d) ratio for Series 1 has the adverse effect in the fracture energy, as computed with the fracture work model 1, as it is observed in Table 5. This suggests that the reinforced matrix with fibers is not capable of arresting the crack growth; this is contrary to what it was observed in Series 2, whereby, by using a larger amount of fibers (i.e., 40 kg/m^3^), the fracture energy increases even if the (a/d) ratio increases. This behavior implies that, due to the presence of a larger amount of fibers, the cracking strength increases by means of a bridging mechanism of the stresses throughout the cracked faces, which can generate a multiple cracking condition in the matrix.

In this manner, the fracture energy obtained in the post-cracking stage will be influenced by the amount of fibers located in the cement matrix and by the capacity of these to transfer stresses in the residual stage, since the extraction process of the steel fibers generates a higher consumption of energy due to the straightening of their hooked end. These will also have the same capacity of diminishing the stress concentration in the upper end of the crack, by limiting the crack propagation and also by limiting the cracking occurrence [36,37,51,52,53,54,55,56,57].

As formerly mentioned, the fracture energy indicates an important contribution of the fibers in the residual stage, given that the average increment of the fracture energy for Series 2 was of more than 100%, for the ratios (a/d) = 0.08 and 0.33, with respect to those results obtained for Series 1. For the ratio (a/d) = 0, the difference is only 57%; this indicates that the notch absence in the specimen does not allow for the efficient behavior of the fibers, thus limiting the increase of energy fracture.

Figure 11a,b show the graphs used for the computation of the fracture energy in the residual stage, for the slope fracture work model (model 2). In these graphs, each set of the tested specimens is represented, along with the initial size of the ligament (the area between the top face of the specimen and the notch tip). It is taken into consideration that the fracture work is proportional to the cracked surface, and such cracked final area is equal to the initial area of the ligament [48].

In the graphs, the necessity of zero energy is considered to fracture beams whose notch goes from top to bottom; this means that the curve is assumed to start at the origin. Furthermore, the slope of the curve represents the energy consumed for the crack to grow a unit in depth. By dividing this slope by the width of the beam (b), it gives as result the fracture energy required.

It is worth mentioning that the results of the fracture energy obtained through this model are relatively close to those obtained by the first model. Such values are represented in Table 6, where GF (1) and GF (2) stand for the fracture energy computed from model 1 and 2, respectively. 

It can be observed that the second model has a direct relation with the variation of the results, since, by obtaining high variation coefficients, the factor R^2^ will also increase. This may lead to a problem to obtain representative results for the behavior of the post-cracking stage of the composite material.

## 5. Conclusions

The results presented in this article are limited to fiber-reinforced concrete with steel fibers, and with the percentage of volume fibers already described. Therefore, experiments with other types of fibers (synthetics and naturals) and with different characteristics are desirable to extend the range of opportunities in the improvement of the post-cracking response of the concrete.

Based on the results, the following conclusions can be drawn:The load at the proportional limit is not affected by the addition of steel fibers, given that the obtained values of load were closed in each of the studied series. This indicates that, for this stage, the performance of the material depends mostly on the cement matrix and the remaining concrete area.The load and the stresses at the proportional limit exhibited an inversely proportional behavior to the notch-to-depth ratio, where an increase in such ratio will result in the concrete to be prone to failure.The increase in the dosage of fibers leads to an improvement in the normal and characteristic residual stresses.For Series 2, the increase of the notch-to-depth ratio (a/d) enhances the normal and characteristic residual performance. For the ratio (a/d) = 0, the increment in the normal residual stress was 61%, while for the ratios (a/d) = 0.08, 0.16, and 0.33, the increment was 157%, 129%, and 86%, respectively. The ratio (a/d) = 0.08 provides the best characteristic residual performance of all the ratios considered.For Series 1, the notch depth of 25 mm, equivalent to a ratio (a/d) of 0.16, was the only ratio that met the minimum residual stress requirements established in international standards.For a low amount of fibers (in this case, 20 kg/m^3^) and for low (a/d) ratios (i.e., a/d < 0.16), it was not possible to reach the minimum classification of residual strength, due to the fact that the specimens were unable to reach the minimum values.The larger classification of residual strength was attained by the ratio of (a/d) = 0.33, which implies that the performance in the post-cracking stage does not depend on the concrete, but rather on the capacity of the fibers to transfer stresses through the cracked faces of the specimen and also on the amount of fibers located in an analyzed section.The fracture energy increased in about 97% (model 1) and in about 35% (model 2), by increasing the volume of fibers from 20 to 40 kg/m^3^. This implies that the steel fibers contribute to improving the residual performance of the composite material.For Series 2, the fracture energy increased even if the (a/d) ratio also increased. The presence of a larger amount of fibers allows the cracking strength to be incremented and to generate a multiple cracking condition in the matrix.The mathematical models used showed similar results, particularly for high contents of steel fibers in the concrete.The results obtained in this research will offer an experimental frame of reference for different ratios (a/d), with respect to the one recommended in the standard, which can facilitate having a criterion of analysis with respect to the residual stresses determined from laboratory tests.

## Figures and Tables

**Figure 1 materials-14-00445-f001:**
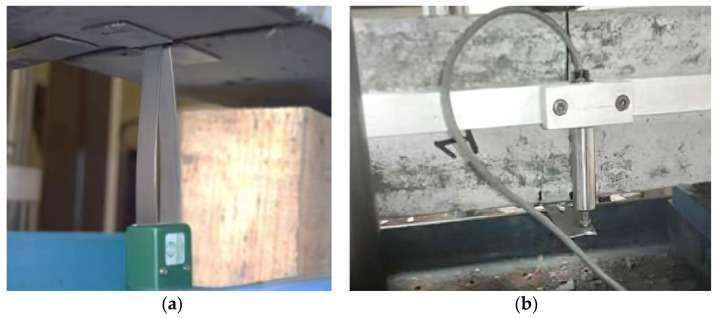
Tensile bending test in prismatic specimens. (**a**) Measurement of the crack-mouth opening by means of extensometer clip-type Epsilon brand. (**b**) Measurement of the deflection at midspan of specimen through a linear variable differential transformer (LVDT), VISHAY brand.

**Figure 2 materials-14-00445-f002:**
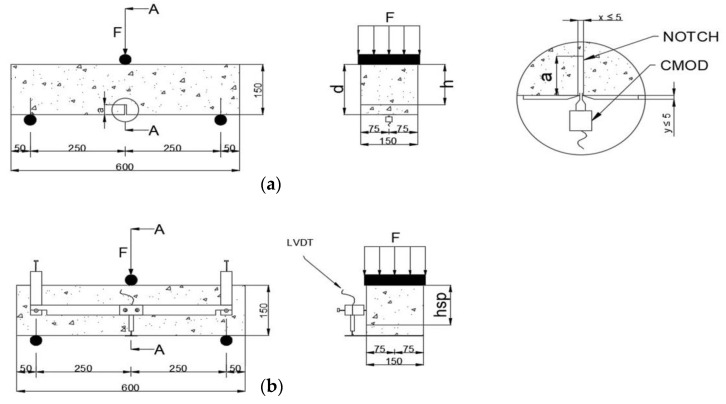
Configuration of the 3-point bending test, dimensions in mm. (**a**) Measurement of the crack-mouth opening. (**b**) Measurement of the displacement at midspan by an LVDT.

**Figure 3 materials-14-00445-f003:**
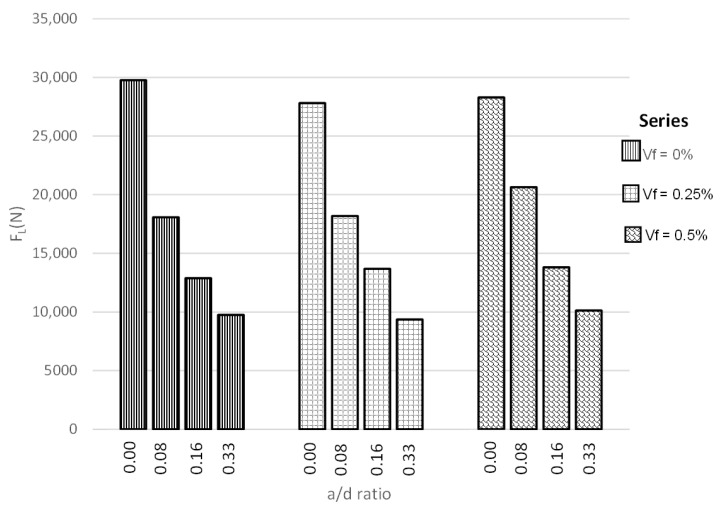
Experimental behavior at the proportional limit.

**Figure 4 materials-14-00445-f004:**
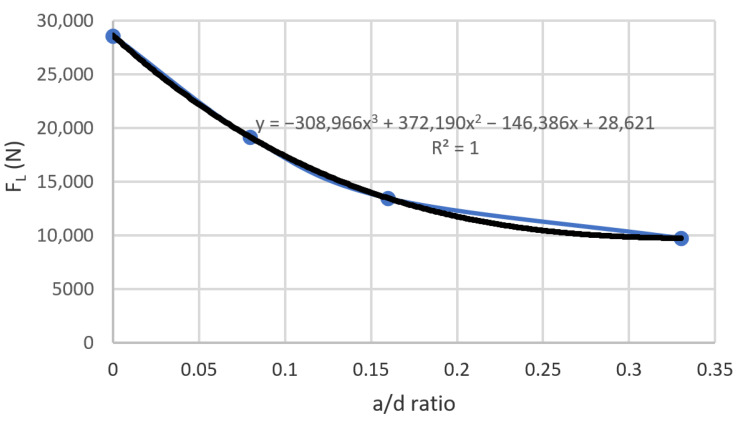
General behavior at the proportional limit.

**Figure 5 materials-14-00445-f005:**
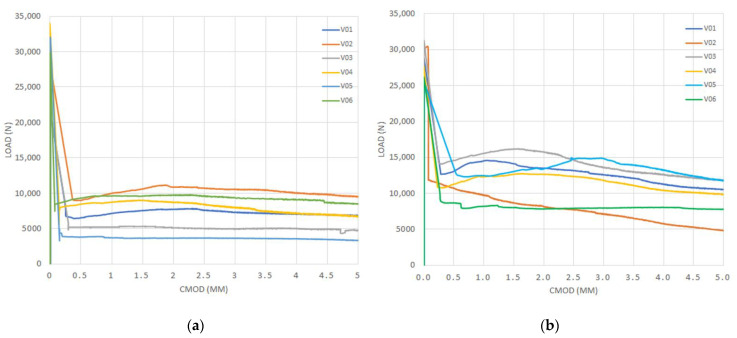
Curves of bending tensile test for the ratio (a/d) = 0: (**a**) Series 1 and (**b**) Series 2.

**Figure 6 materials-14-00445-f006:**
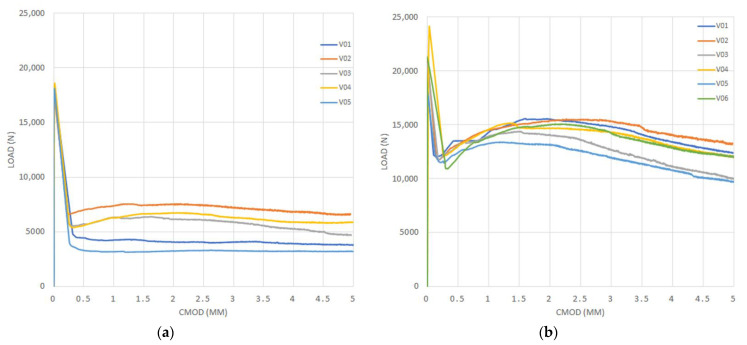
Curves of bending tensile test for the ratio (a/d) = 0.08: (**a**) Series 1 and (**b**) Series 2.

**Figure 7 materials-14-00445-f007:**
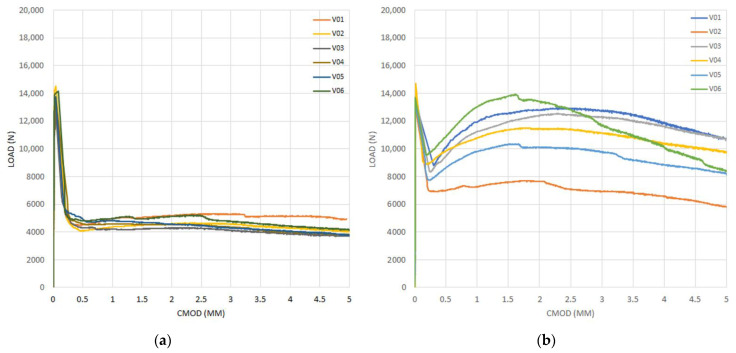
Curves of bending tensile test for the ratio (a/d) = 0.16, (**a**) Series 1, (**b**) Series 2.

**Figure 8 materials-14-00445-f008:**
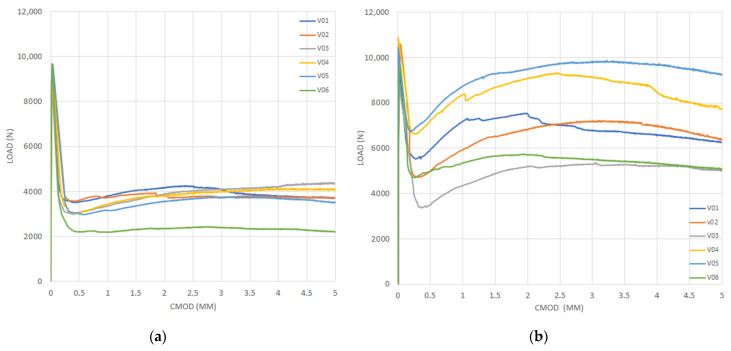
Curves of bending tensile test for the ratio (a/d) = 0.33: (**a**) Series 1 and (**b**) Series 2.

**Figure 9 materials-14-00445-f009:**
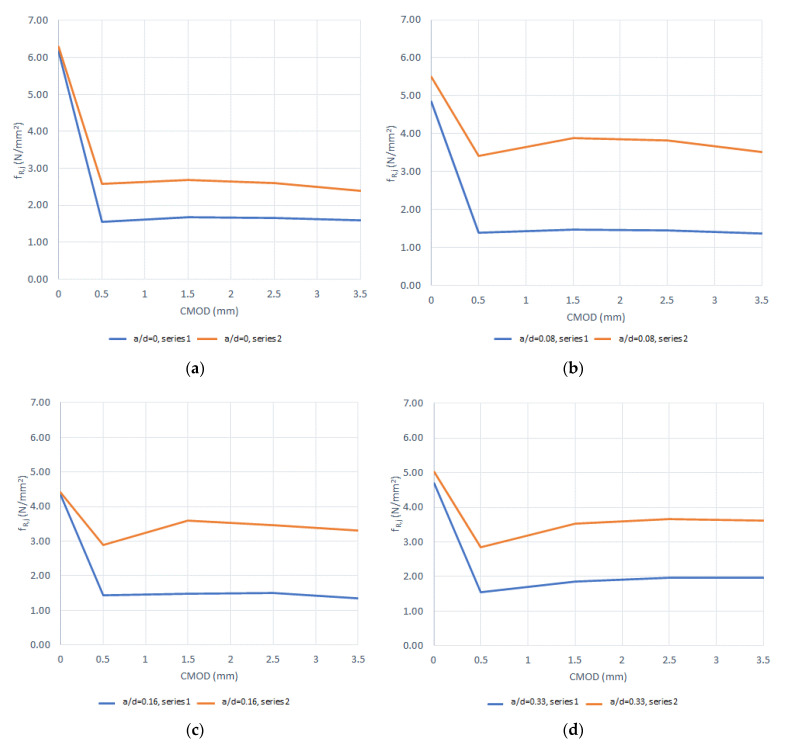
Residual normal stresses for both series with fibers and for each of the studied (a/d) ratios (**a**) 0, (**b**) 0.08, (**c**) 0.16, and (**d**) 0.33.

**Figure 10 materials-14-00445-f010:**
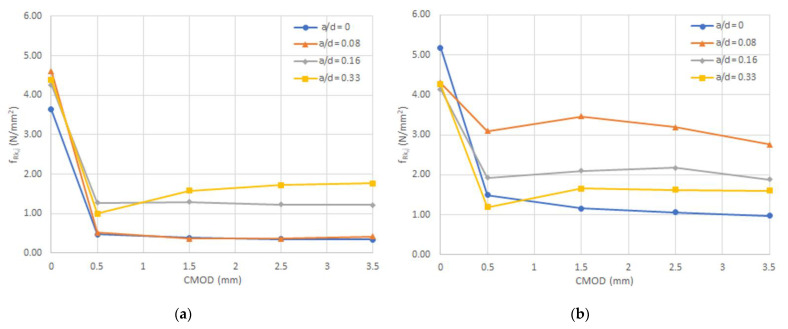
Characteristic residual strength for each (a/d) ratio: (**a**) Series 1 and (**b**) Series 2.

**Figure 11 materials-14-00445-f011:**
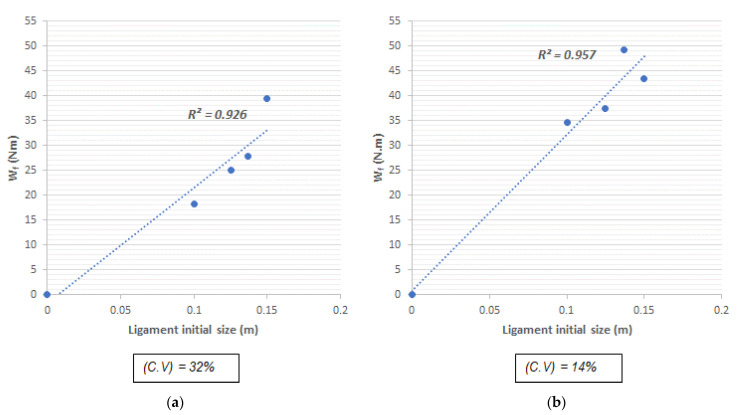
Curve fit for the computation of the fracture energy by the slope fracture work model 2: (**a**) Series 1 and (**b**) Series 2.

**Table 1 materials-14-00445-t001:** Mixture composition.

Material	Mix		
	S-1	S-2	S-3
Cement (kg/m^3^)	350	350	350
Additive (ml/kg)	1.9	1.9	1.9
Gravel (M.S. 19 mm) (kg/m^3^)	810	810	810
Sand (kg/m^3^)	1027	1020	1014
Water (kg/m^3^)	193	193	193
Fiber (kg/m^3^)	0	20	40
Air content (%)	1.8	2.5	2.8
Slump (mm)	130	115	105
Vebe time (s)	6	7	9

**Table 2 materials-14-00445-t002:** Factor k_x_ as a function of the number of tested specimens.

n	1	2	3	4	5	6	8	10
k_xN_	2.31	2.01	1.89	1.83	1.8	1.77	1.74	1.72
k_xn_			3.37	2.63	2.33	2.18	2	1.92

**Table 3 materials-14-00445-t003:** Computed stresses at the proportional limit (N/mm^2^).

	Series											
	Reference				S-1				S-2			
					Ratio (a/d)							
	0	0.08	0.16	0.33	0	0.08	0.16	0.33	0	0.08	0.16	0.33
**V01**	6.84	4.70	4.13	4.82	4.75	4.84	4.34	4.65	6.41	5.04	4.35	4.88
**V02**	6.39	4.83	4.05	4.88	6.26	4.69	4.64	4.84	6.74	5.58	4.28	5.27
**V03**	7.98	4.80	4.09	4.98	4.80	4.90	4.32	4.55	6.95	5.50	4.45	4.51
**V04**	7.48	4.73	4.34	4.89	7.51	4.96	4.33	4.53	6.14	6.41	4.59	5.43
**V05**	8.40	4.95	4.17	4.95	7.10	4.83	4.40	4.81	5.67	4.83	4.29	5.22
**V06**	7.92	4.92	4.03	4.73	6.64	-	4.31	4.82	5.81	5.63	4.45	4.79
**S**	0.76	0.10	0.11	0.09	1.17	0.10	0.04	0.14	0.51	0.55	0.12	0.35
**Ẋ**	7.50	4.82	4.13	4.87	6.18	4.84	4.34	4.70	6.29	5.50	4.40	5.02
**CV (%)**	10.16	2.04	2.73	1.85	18.87	2.04	0.85	3.01	8.09	9.98	2.74	6.93

**Table 4 materials-14-00445-t004:** Classification of the residual stress for Series 1 and 2.

**Ratio (a/d)**	**CMOD (0.5 mm)**	**CMOD (2.5 mm)**	**f_Rk,3_/f_Rk,1_**	**Classification**	**Post Cracking Response**
	**f_Rk,1_ (N/mm^2^)**	**f_Rk,3_ (N/mm^2^)**			
**Series 1**					
**0**	0.48	0.36	0.74		
**0.08**	0.52	0.37	0.72		
**0.16**	1.27	1.23	0.97	1c	Perfectly plastic
**0.33**	1	1.72	1.73	1e	Hardening
**Series 2**					
**0**	1.5	1.07	0.71		
**0.08**	3.09	3.19	1.03	3c	Perfectly plastic
**0.16**	1.93	2.17	1.13	1d	Soft hardening
**0.33**	1.19	1.63	1.36	1e	Hardening

**Table 5 materials-14-00445-t005:** Computed fracture energy with model 1 [47] for the specimens reinforced with fibers (Series 1 and Series 2).

**Series 1**					**Series 2**			
**Specimen**	**GF (N/m)**	**Ẋ (N/m)**	**S (N/m)**	**CV (%)**	**GF (N/m)**	**Ẋ** **(N/m)**	**S (N/m)**	**CV (%)**
**(a/d) = 0**								
**V01**	1263.87	1258.56	398.11	31	2226.93	1976.32	439.29	22
**V02**	1797.68				1460.97			
**V03**	959.26				2450.16			
**V04**	1441.45				1998.46			
**V05**	714.19				2299.28			
**V06**	1554.9				1422.09			
**(a/d) = 0.08**								
**V01**	873.39	1107.31	304.31	27	2541.02	2451.36	136.96	6
**V02**	1350.57				2571.27			
**V03**	1400.2				2360.64			
**V04**	1206.25				2532.56			
**V05**	706.12				2216.17			
**V06**	-				2486.53			
**(a/d) = 0.16**								
**V01**	1654.08	1159.33	277.34	24	2261.91	2019.59	336.09	17
**V02**	1099.61				1432.05			
**V03**	941.06				2185.08			
**V04**	946.92				2069.61			
**V05**	1009.75				1836.91			
**V06**	1304.58				2332.01			
**(a/d) = 0.33**								
**V01**	1022.07	924.6	130.75	14	2328.8	2383.8	555.05	23
**V02**	999.66				2283.87			
**V03**	960.7				1739.47			
**V04**	966.51				2905.11			
**V05**	933.33				3155.94			
**V06**	665.36				1889.57			

**Table 6 materials-14-00445-t006:** Fracture energy (GF) computed through the studied models [58].

(GF) (N/m)			ΔGF (N/m)	ΔGF (%)
Series	GF (1)	GF (2)		
1	1119.95	1549.2	429.25	38.33
2	2207.77	2089.07	118.70	5.38

## Data Availability

Publicly available datasets were analyzed in this study. This data can be found here: [http://eprints.uanl.mx/16177/].

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
