# Peer review of "Effect of the Notch-to-Depth Ratio on the Post-Cracking Behavior of Steel-Fiber-Reinforced Concrete"

_materials, 2021, doi:10.3390/ma14020445_

Round 1

Reviewer 1 Report

The authors intended to determine the effect of the addition of the steel fibers into the performance of the post-cracking stage by studying four notch to depth ratio. The methodology was described properly and the results are meaningful to some extent. However, there are some problems existing in this paper which the authors must pay attention to deal with.

  1. The background and significance of this study should be highlighted in the abstract.
  2. In the Introduction, the literature review is not comprehensive. Please study the previous research on this topic in the past 3 years. Pay attention to some related literatures from Europe and China.
  3. Some expressions in the manuscript are not clear. An English native speaker is suggested to carefully proofread it again. For example, in the abstract “This is evaluated through 72 bending tests using plain concrete as control and volume fibers of 0.25% and 0.50%.” is suggested to be “This was evaluated through 72 bending tests using plain concrete as control and volume fibers of 0.25% and 0.50%.”
  4. Many mistakes can be found as well. Please carefully check through the manuscript. For example, in the introduction “However, a large influence on tensile strength [14-16], and a substantial influence on impact loading and residual strength tests [17-20], have been shown Steel fibers maybe suitable for reinforcing concrete, since provide an energy dissipation mechanism and control the crack propagation in residual stages more efficiently than plain concrete, with the latter as one of the main advantages in the use of this reinforcing material;” “For the computation of the residual strength, international codes and standards have proposed 3-point bending tests on notched beam specimens at midspan, to control the crack development, by establishing a notch to depth ratio (i.e., a/d) of 0.16. [Ref missing].” Equation (4) was ordered as equation (1).
  5. The purpose of this research should be proposed in the end of introduction.
  6. Most figures are of low quality. Different curves are difficult to be distinguished in the Figures 5-8.
  7. Some limitation and outlook is suggested to be mentioned in the last chapter.

Reviewer 2 Report

Comments

This paper investigated the effect of the notch to depth ratio on the post-cracking behaviour of steel fiber reinforced concrete. The outcome is interesting for readers. However, there are several aspects that need to be improved. The reviewer can only recommend for publication if the author satisfactorily address the following comments in the revised version.

  1. The illustration of notch-to-depth ratio need to be provided for better understanding.
  2. What are the effects of notch-to-depth ratio on fracture energy, normal residual stress, and characteristic residual stress? This can be one dot point in conclusion section.
  3. The novelty of the study should be highlighted in the end of introduction section. How this study is different from the published study in literature?
  4. How the outcome of this study will benefit researchers and end users? This need to be highlighted in introduction or end of conclusion.
  5. The failure mechanism of the specimens should be discussed more clearly.
  6. The background on a/d ratio and fibre reinforced concrete need to be strongly highlighted in introduction section. It was found that a/d ratio has strong influence on failure behaviour [Ref: Effect of beam orientation on the static behaviour of phenolic core sandwich composites with different shear span-to-depth ratios]. Moreover, the fibres can improve the overall structural capacity [Ref: Performances, challenges and opportunities in strengthening reinforced concrete structures by using FRPs-A state-of-the-art review]. Suggest to include them in introduction section with proper citations.

I would be happy to see the revised version to understand how these comments are being addressed.

Reviewer 3 Report

Presented article describes effects of steel fiber reinforcement of concrete on mechanical behaviour of concrete.

The writing itself does not flow very well. Some issues needed second iteration of reading to be understood, hence the quality should be improved for this level of work. Quality of photos and graphs also should be improved.

Referenced articles are quite outdated. In my opinion, the introduction of presented article should implement the actual researches in the field. Review of the literature should indicate the undertaken, new problem, that is being researched.

L. 2-5 and further: "However, concrete barely possesses tensile strength and it is susceptible to cracking, which leads to a reduction of its service life, since, once cracks start developing, concrete lacks mechanical strength and fails suddenly [1-8]." Sentence is very long and is referenced with 8 literature positiones. It would be advisable to point out precisely what the quoted items contribute to the current state of research.

Figure 1. It is hard to perceive what is presented on the photos. I would suggest to improve its quality.

Figure 2: What does it mean Muesca?

Equation (4) is marked as equation (1).

There is many mistaken paragraphs, fe. in 9'th line form the bottom on page No 6.

All used abbrevations should be explained.

Text above Figure No 5 is finished in the middle of the sentence.

Part of the graphs is outscaled, fe. presented on figures 6, 7 and 8. It would be easier to understand them if the y axis would be limitted to the values used by the data. For example Figure 7 a shows the load of maximum 1500 N, and Y axis is widen up till 4000 N.

Page No 12 - mistakes in paragraphs and justification.

Many to long sentences which are hard to understand, fe. page 14:

"In the
graphs, the necessity of zero energy for fracturing completely notched beams is considered, which leads to the curve to start at the origin; the slope of the curve represents the energy consumed for the crack to grow a unit in depth."

Yours Sincerely,

Reviewer.

Round 2

Reviewer 1 Report

I am pleased that the questions were carefully answered by authors, and the suggestions were all considered by the authors according to the feedback. However, it is suggested that the manuscript should be proofread again and some modifications should be made.

Reviewer 2 Report

I have no further comments.

Reviewer 3 Report

Dear Authors

I would suggest you to try to improve the quality of the pictures, especially presented in figure No 2. It seems to be a low-resolution one.

All my previous notes and suggestions have been answered and the article has been properly corrected. In my opinion, it should be published in Materials after Minor revision pointed above.

Congratulations on presented work!